# Identification of MYEOV-Associated Gene Network as a Potential Therapeutic Target in Pancreatic Cancer

**DOI:** 10.3390/cancers14215439

**Published:** 2022-11-04

**Authors:** Yu Chen, Jialun Wang, Qiyuan Guo, Xihan Li, Xiaoping Zou

**Affiliations:** Department of Gastroenterology, Affiliated Nanjing Drum Tower Hospital, Medical School of Nanjing University, Nanjing 210008, China

**Keywords:** pancreatic cancer, MYEOV, KRAS, ceRNA, immune infiltration

## Abstract

**Simple Summary:**

Exploring the key molecular regulatory mechanism of pancreatic cancer is crucial to investigate the therapeutic strategies for pancreatic cancer. To precisely identify key pancreatic cancer genes, we used a multiple analyses integrated approach. We performed gene association analysis on the key gene MYEOV we discovered, which led to a molecular network with MYEOV as the core. The genes in this molecular network could work together to influence the development of pancreatic cancer, possibly through protumor immune cell infiltration. It is the first report to focus on the possibility that MYEOV may act as a ceRNA to form an interaction network with some pancreatic cancer-related genes such as KRAS and serve as a strategic therapeutic target of pancreatic cancer treatment.

**Abstract:**

The molecular mechanism that promotes pancreatic cancer remains unclear, so it is important to find the molecular network of important genes related to pancreatic cancer. To find the key molecule of pancreatic cancer, differential gene expression analyses were analyzed by the Deseq2 package, edgeR package, and limma-voom package, respectively. Pancreatic cancer survival-related genes were analyzed by COX survival analysis. Finally, we integrated the results to obtain the significantly differentially expressed gene, MYEOV (myeloma overexpressed gene), most strongly related to survival in pancreatic cancer. Experimental verification by qRT-PCR confirmed that transcription levels of MYEOV mRNA markedly increased in pancreatic cancer cells relative to normal human pancreatic ductal epithelial cells (HPDE). Through the comprehensive analysis of multiple databases, we constructed a molecular network centered on MYEOV and found specific links between molecules in this network and tumor-associated immune cells. It was noted that MYEOV could serve as a ceRNA by producing molecular sponging effects on hsa-miR-103a-3p and hsa-miR-107, thus affecting the role of GPRC5A, SERPINB5, EGFR, KRAS, EIF4G2, and PDCD4 on pancreatic cancer progression. Besides, we also identified that infiltrated immune cells are potential mediators for the molecules in the MYEOV-related network to promote pancreatic cancer progression. It is the first report to focus on the possibility that MYEOV may act as a competing endogenous RNA (ceRNA) to form an interactive network with some pancreatic cancer-related genes such as KRAS and serve as a key therapeutic target of pancreatic cancer treatment.

## 1. Introduction

Cancers, as we all know, have become a health killer worldwide, with high morbidity and mortality [1]. Among them, pancreatic cancer is a malignancy with its mortality rate slowly increasing in the past years. It is expected to become the second leading cause of cancer deaths in the United States within the next 20 to 30 years [2]. GLOBOCAN estimates of the global incidence of pancreatic cancer show an increasing trend, with 495,773 new confirmed cases in 2020 and associated deaths of 466,003 cases [3]. Pancreatic cancer usually presents few obvious symptoms until it progresses to advanced stages and is often not easily diagnosed early [4]. Although there have been many research advances for treating pancreatic cancer in recent years, many limitations remain [5]. Although strategies such as radiotherapy techniques and targeted therapeutic agents have improved the therapeutic efficacy of pancreatic cancer to some extent, the five-year survival rate for early-stage patients is still low [6]. Therefore, studying the molecular mechanism of pancreatic cancer to better understand the cause of pancreatic cancer development is important for both the diagnosis and treatment of pancreatic cancer [7,8].

Various studies have performed bioinformatic analyses for molecules specific to pancreatic cancer, but most of them fail to provide truly effective targets for the diagnosis and treatment of pancreatic cancer [9]. To identify genes critical for pancreatic cancer, we filtered and focused on the candidate gene MYEOV by combining diverse differential gene expression analyses and COX survival analysis for pancreatic cancer and normal pancreatic tissues. MYEOV has been found to have transcripts that do not match its protein expression levels in some cancers, as dramatically highly-expressed MYEOV transcripts cannot be translated to mature proteins in these cancer cells [10]. It may be linked to its 5′UTR sequence that can exert inhibitory effects on its protein synthesis. This implies that MYEOV may play a significant role in the form of mRNA instead of protein. A study has revealed that MYEOV functions as a competing endogenous RNA (ceRNA) to promote non-small cell lung cancer through the TGF-β signaling pathway [11]. One important role of ceRNA is to act as a molecular sponge for miRNAs, thus competing for other targets of the shared miRNAs [12]. According to the hypothesis of ceRNA, they can function as a molecular sponge to isolate miRNAs and liberate their mRNA targets from inhibition, thus promoting the development of the pro-cancer effects produced by these downstream mRNA targets [13]. Our results also indicated that the effect of MYEOV on pancreatic cancer progression could be based on its ceRNA role.

The objective of this study was to uncover the molecular regulatory network associated with the mechanism of pancreatic cancer development by uncovering key pancreatic cancer-related genes. Given the specificity of the MYEOV gene we identified, we innovatively explored the regulatory role of MYEOV as a ceRNA for various pancreatic cancer key genes. We also identified the key microRNAs (miRNAs) among these genes and revealed the key crosstalk relationships of the MYEOV-related molecular regulatory network. Through tumor-associated immune infiltration analysis, we identified that these pancreatic cancer-associated genes might mediate the development of pancreatic cancer by tumor-associated immune infiltrating cells. The main results of this study provide important possibilities to study the generation and malignant biological development of pancreatic cancer and also provide new targets for early diagnosis and precise treatment of pancreatic cancer in the future.

## 2. Materials and Methods

### 2.1. Data Collection and Analysis

RNA sequencing data of pancreatic cancer were downloaded from The Cancer Genome Atlas (TCGA) database (https://cancergenome.nih.gov/abouttcga/overview (accessed on 30 April 2022)), and differential analysis was performed using the Deseq2 package, edgeR package, and limma-voom package of R software (version 3.6.3). Batch survival analysis was conducted using the COX analysis method in the survival R package.

### 2.2. Gene Differential Expression Validation

Differential expression analyses of relevant genes were validated using the GEO database (https://www.ncbi.nlm.nih.gov/geo/ (accessed on 30 April 2022)) and the GEPIA database (http://gepia.cancer-pku.cn/ (accessed on 30 April 2022)). Log2(TPM + 1) was used for the log scale. Analysis of differential gene expression was performed by data from the GTEx and TCGA portals. The correlation of MYEOV with tumor grade was analyzed by the TISIDB database (http://cis.hku.hk/TISIDB (accessed on 30 April 2022)). TISIDB is a database that integrates multiple heterogeneous data types, including but not restricted to TCGA.

### 2.3. Relative mRNA Levels Measurements

Pancreatic cancer tissues and normal tissues adjacent to pancreatic cancer were harvested from patients who underwent surgery in Nanjing Drum Tower Hospital from July 2017 to July 2019. All samples were histologically reviewed by two pathologists. None of the patients received radiotherapy or chemotherapy before surgery. All specimens were immediately frozen in liquid nitrogen within 5 min after resection and then stably sorted at −80 °C. Clinical samples were collected from patients after written informed consent was obtained. Human pancreatic cancer cells (HPAC, Bxpc-3, Panc1, CFPAC) were obtained from the American Type Culture Collection (ATCC), and HPDE cells were a gift from the Technical University of Munich, Germany. All cell lines were grown in the recommended culture media and maintained at 37 °C in 5% CO_2_. 

Quantitative real-time RT-PCR (qRT-PCR) was used to validate relative mRNA levels. Total RNA was extracted from the cultured cells by the TRIzol method (Invitrogen, Waltham, MA, USA). According to the manufacturer’s protocols (Vazyme, Nanjing, China), cDNA was synthesized by reverse transcription and then used as a substrate of qRT-PCR. GAPDH expression was employed as an internal control for mRNA. The primers used for qRT-PCR are given in Appendix A.

### 2.4. The Construction, Transfection, and Infection of Small Interfering RNA (siRNA)

The siRNAs against human MYEOV (siMYEOV-1, F-5′-UCUGUGACAU CAUGCUGCATT-3′, R-5′-UGCAGCAUGAUGUCACAGATT-3′, siMYEOV-2, F-5′-UUUCUGUGACAUCAUGCUGTT-3′, R-5′-CAGCAUGAUGUCACAGAAATT-3′, siControl, F-5′-UUCUCCGAACGUGUCACGUTT-3′, and R-5′-ACGUGACACGUUCGGAGAATT-3′) were purchased from Tsingke. The siRNAs were transfected into cells using Lipofectamine 3000 (Invitrogen) according to the manufacturer’s protocol.

### 2.5. CCK-8 Assay and Cell Colony Formation Assay

For the CCK-8 assay, cells were resuspended at a cell density of 5 × 103 cells/well by a complete culture medium in the 96-well plates. After 48 h of culturing at 37 °C with 5% CO_2_, 10 μL CCK-8 solution (Dojindo, Rockville, MD, USA) was spiked into each well, and the plates were incubated at 37 °C for 2 h. After that, the absorbance of each well at 450 nm was measured with a Tecan Infinite M1000 microplate reader.

For the cell colony formation assay, 500 cells were seeded into each well of a 6-well plate and incubated at 37 °C with 5% CO_2_ for about 14 days. Then the plate was gently washed, fixed with methanol, and stained with crystal violet.

### 2.6. Western Blotting

Samples were homogenized in RIPA buffer, sonicated, and prepared for Western blotting. Protein samples were separated by electrophoresis on 10% SDS-PAGE gels before semi-dry transfer to the PVDF membrane. MYEOV, GPRC5A, KRAS, and α-tubulin were detected using antibodies purchased from Proteintech. Original WB cam be found at File S1.

### 2.7. Gene-Related Survival Validation

The correlation of MYEOV with OS in cancers was analyzed using the log-rank test by the TISIDB database. GSCA database (http://bioinfo.life.hust.edu.cn/GSCA/#/ (accessed on 30 April 2022)) was used to analyze the overall survival (OS), progression-free interval (PFI), disease-free interval (DFI), and disease-specific survival (DSS) of MYEOV. RNA-sequencing data of 269 Australian patients with pancreatic cancer were downloaded from the ICGC dataset (https://dcc.icgc.org/releases/current/Projects (accessed on 30 April 2022)), and the log-rank test was used to compare differences in OS. The OS and disease-free survival (DFS) of related genes were analyzed using the GEPIA database.

### 2.8. Gene Co-Expression Analysis

Statistical analysis of gene co-expression was performed based on the Pearson correlation coefficient using the LinkedOmics database (http://linkedomics.org/login.php (accessed on 30 April 2022)). Gene correlation analysis was also validated by the TIMER2 database (http://timer.cistrome.org/ (accessed on 30 April 2022)) and the GEPIA database.

### 2.9. Alteration Prediction

The cBioPortal database (https://www.cbioportal.org/ (accessed on 30 April 2022)) was used for MYEOV and GPRC5A alteration-related genes prediction. A total of 1174 samples of pancreatic cancer in the cBioPortal database were analyzed.

### 2.10. PPI Analysis

Integrated network analysis of molecules associated with MYEOV, GPRC5A, and KRAS was performed by the String database (https://cn.string-db.org/ (accessed on 30 April 2022)). The prediction was made on the basis of curated databases, experimentally determined gene co-occurrence or text mining, with a minimum required interaction score of 0.4.

### 2.11. Gene-Related Immune Infiltration Analysis

Infiltration of immune cells associated with gene expression in pancreatic cancer was analyzed by the GSCA database. The immunogenomic analysis we performed in the GSCA database was for 24 immune cells, calculated by the ImmuCellAI algorithm.

### 2.12. miRNA Analysis

The target miRNAs of related genes were predicted by searching the ENCORI database (https://starbase.sysu.edu.cn/ (accessed on 30 April 2022)), and key miRNAs were obtained by taking intersections using the Jvenn database (http://jvenn.toulouse.inra.fr/app/index.html (accessed on 30 April 2022)). The CancerMIRNome database (http://bioinfo.jialab-ucr.org/CancerMIRNome/ (accessed on 30 April 2022)) and KM-Plotter website (http://kmplot.com/ (accessed on 30 April 2022)) were used to verify the differential expressions of miRNAs in tumors and their impacts on survival. Target gene enrichment analyses of miRNAs were performed using the miRPath database (http://www.microrna.gr/miRPathv3/ (accessed on 30 April 2022)).

### 2.13. Dual-Luciferase Reporter Assay

The MYEOV/KRAS wild-type (WT) and mutant (MUT) 3′ UTR were created and cloned to the pmirGLO Dual-Luciferase Reporter Vector (Promega, Madison, WI, USA) by Qingke. The miRNA mimics were also purchased from Qingke. CFPAC cells were seeded in 6-well. Luciferase plasmids and hsa-miR-103a-3p/hsa-miR-107 mimics or negative-control (NC) were co-transfected into CFPAC cells with Lipo 2000 Transfection Reagent (Invitrogen). The luciferase activity was assessed using the dual-luciferase reporter kit (Promega) and a Tecan Infinite M1000 microplate reader.

### 2.14. Statistical Analyses

All statistical analyses were carried out using Graph Pad Prism (version 9.0) and R software. *p*-value < 0.05 indicates the significance of the correlation.

## 3. Results

### 3.1. MYEOV as a Key Upregulated Gene Closely Related to Prognosis in Pancreatic Cancer

To investigate the cause of pancreatic cancer development at the molecular level, we performed differential analysis by calling PAAD-related data from the TCGA database and using the Deseq2 package, edgeR package, and limma-voom package in R language, respectively. We took the intersection of genes significantly upregulated in pancreatic cancer from the three analyses and obtained 716 upregulated genes (Figure 1A). We also performed COX survival analysis on the PAAD data from the TCGA database and obtained 1836 genes significantly associated with survival. After intersecting these 1836 genes with the 716 upregulated genes, we obtained MYEOV, the most significant gene with differential survival among them (*p* = 3.6 × 10^−5^). To validate this result, we used the data combined by GSE107610 and GSE16515 datasets from the GEO database. We confirmed that MYEOV expression was dramatically increased in pancreatic cancer tissues relative to normal tissues (Figure 1B). Through experiments, we demonstrated the differential expression of MYEVO in HPDE cells and four pancreatic cancer cells. Considerably increased MYEOV expression was found in all four pancreatic cancer cells relative to HPDE cells (Figure 1C). In the TISIDB database, we further verified that MYEOV was associated with higher grade and shorter OS in pancreatic cancer, while there was no significant association with the higher stage (Figure 1D and Appendix A). We validated the survival differences between patients with high and low MYEOV expression using log-rank tests in pancreatic cancer patients by the GSCA database. It was found that high expressed MYEOV was significantly associated with shorter survival in pancreatic cancer patients, both in OS (*p* = 2.3 × 10^−6^), DSS (*p* = 3.5 × 10^−7^), DFI (*p* = 0.063), and PFS (*p* = 1.7 × 10^−4^) (Figure 1E–H). The results of survival analysis in the ICGC database for pancreatic cancer patients also showed that high MYEOV expression was indeed significantly associated with shorter OS (Appendix A). To investigate the function of MYEOV in pancreatic cancer cells, we effectively knocked down MYEOV by siRNAs in CFPAC. CCK-8 assays and colony formation assays were carried out to explore the effects of MYEOV on pancreatic cancer cell proliferation. These experiments showed that the knockdown of MYEOV drastically inhibited pancreatic cancer cell proliferation (Figure 1I,J).

### 3.2. Gene Co-Expression Analysis of MYEOV in Pancreatic Cancer

To deeply investigate the role of MYEOV on the mechanism of pancreatic carcinogenesis, we analyzed the genes associated with MYEOV in PAAD by RNA high-throughput sequencing data from the LinkedOmics database. We obtained the most strongly associated gene, GPRC5A, which was significantly more associated with MYEOV than other genes (Figure 2A,B). GPRC5A is also an under-recognized pancreatic cancer-associated gene and showed high expression levels in multiple cancers (Appendix A). We performed alteration-associated gene predictions for MYEOV and GPRC5A by using the cBioPortal database. It was found that the KRAS gene was in the first position for both MYEOV and GPRC5A, suggesting that KRAS may have an important association with them (Figure 2C). Activating mutations in the KRAS gene are widespread in more than 90% of pancreatic cancers and are considered to be an essential cause of pancreatic carcinogenesis [14]. To further discover whether KRAS is associated with these two genes, we performed a gene association analysis by the Timer2 database and the GEPIA database. The results showed that there was a strong positive association between MYEOV and GPRC5A (*p* = 3.44 × 10^−19^), a strong positive association between GPRC5A and KRAS (*p* = 5.15 × 10^−17^), and a significant positive association between MYEOV and KRAS (*p* = 5.72 × 10^−5^), indicating that all three may exist in one molecular network (Figure 2D–F). To better validate the above findings, we validated three gene expressions in human tissues and human cells, respectively. The protein expression levels of MYEOV, GPRC5A, and KRAS showed a consistent trend in pancreatic cancer tissues and normal tissues adjacent to pancreatic cancer from three pancreatic cancer patients. They all showed relatively high levels in tumor tissues relative to normal tissues (Figure 2G). To further verify the regulatory relationship of MYEOV on GPRC5A and KRAS, we knocked down MYEOV by siRNAs in CFPAC cells. We found that GPRC5A and KRAS showed corresponding decreasing trends, and GPRC5A changed more obviously with MYEOV (Figure 2H and Appendix A).

### 3.3. PPI Analysis and Molecular Regulatory Network of MYEOV

To better investigate how MYEOV, GPRC5A, and KRAS act together in pancreatic cancer, we utilized the String database, a dedicated database for studying protein interaction networks. Through the String database, we found that MYEOV, GPRC5A, and KRAS were able to have interactions with genes such as SERPINB5, EGFR, EIF4G2, and PDCD4 (Figure 3A). Association analysis by the Timer2 database and the GEPIA database also showed that MYEOV was significantly associated with SERPINB5, EGFR, and EIF4G2 positively, while there was a significant negative association with PDCD4 (Figure 3B–E). Despite the strong association of these genes with MYEOV, no studies have been conducted to fully clarify the precise interactions of MYEOV with these genes. By searching The Human Protein Atlas database, we found that the subcellular localizations of the proteins encoded by these genes differed significantly. MYEOV protein is mainly localized in nucleoplasm and vesicles. GPRC5A protein is mainly localized in vesicles and plasma membrane, and SERPINB5 protein is mainly localized in vesicles and plasma membrane (Figure 3F). EGFR protein is mainly detected in the plasma membrane and cell junctions and is predicted to be secreted (Figure 3F). EIF4G2 is localized to the cytosol, and PDCD4 is localized to the nucleoplasm (Figure 3F). KRAS is localized to the inner cell membrane and is a GTPase that can be activated by extracellular upstream signals, thus regulating intracellular downstream signals [15]. At the protein level, it seems impossible to find a perfect pathway to explain how these molecules interact with each other, suggesting that there seem to be some other associations that have been uncovered.

### 3.4. MYEOV as ceRNA for the Regulation of Multiple Genes

A study has found that MYEOV can act as a ceRNA in tumors in the form of mRNA that binds to specific miRNAs, thus isolating the binding effect of miRNAs on target mRNAs [11]. Through the analysis of the ENCORI database, we found that MYEOV can interact with multiple miRNAs as a circRNA, which may be crucial for MYEOV to be associated with other genes. We searched for miRNAs bound to GPRC5A, SERPINB5, KRAS, EGFR, EIF4G2, and PDCD4 and took intersections respectively to find hsa-miR-103a-3p, hsa-miR-107, hsa-miR-186-5p, hsa-miR-524-5p, and hsa-miR-520d-5p (Figure 4A). In the ENCORI database, we analyzed all miRNAs that interacted with MYEOV as a circRNA and tried to see if they could overlap with the miRNAs that interacted with these six genes. We were surprised to find two miRNAs, hsa-miR-103a-3p and hsa-miR-107, in both sets of miRNAs. The LinkedOmics database exploration revealed that hsa-miR-103a-3p was highly positively correlated with hsa-miR-107 (Figure 4B,C). Their action sites with MYEOV are completely consistent, which can also be seen in the sites analyses of GPRC5A, SERPINB5, KRAS, EGFR, EIF4G2, and PDCD4 (Figure 4D). All these results imply that these two miRNAs may play an important regulatory role in the expression of these genes. To clarify if these two miRNAs are important nodes in the molecular network related to pancreatic carcinogenesis and development, we studied the relationship between miRNA levels and pancreatic cancer by the CancerMIRNome database and GEO database analysis. We confirmed that both hsa-miR-103a-3p and hsa-miR-107 were relatively elevated in pancreatic cancer relative to normal pancreatic tissue, with hsa-miR-107 being more significantly elevated (*p* = 0.016), and high expression of the two miRNAs in pancreatic cancer tissues were both associated with better survival prognosis (Figure 4E). In addition, our analyses in other cancers also found that hsa-miR-103a-3p and hsa-miR-107 showed similar levels and impacts on survival (Appendix A). The miRPath database searching results of target gene pathway enrichment analyses for hsa-miR-103a-3p and hsa-miR-107 also suggested an important association of these two miRNAs with the development of various cancers, such as pancreatic cancer (Appendix A). The ratios of firefly fluorescence (LUC) to Renilla fluorescence (REN) were significantly decreased in CFPAC cells co-transfected with hsa-miR-103a-3p/hsa-miR-107 mimics and pmirGLO-MYEOV-WT or pmirGLO-KRAS-WT compared with the negative control (Figure 4F,G). Interestingly, the LUC to REN ratio did not change significantly when the CFPAC cells were co-transfected with hsa-miR-103a-3p/hsa-miR-107 mimics and pmirGLO-MYEOV-MUT or pmirGLO-KRAS-MUT. Therefore, MYEOV and KRAS both have the binding specificity with hsa-miR-103a-3p and hsa-miR-107.

### 3.5. Analysis of the Role of Genes in MYEOV Regulatory Network on Pancreatic Cancer

To investigate whether MYEOV acts as ceRNA to exert certain regulatory effects on genes such as GPRC5A, SERPINB5, KRAS, EGFR, EIF4G2, and PDCD4, we analyzed the differential expression of these genes in pancreatic cancer and normal pancreatic tissues by the GEPIA database. It was found that MYEOV, GPRC5A, SERPINB5, KRAS, EGFR, and EIF4G2 were all significantly elevated in pancreatic cancer tissues, and the expression trends of GPRC5A, SERPINB5, KRAS, EGFR, and EIF4G2 in different stages were nearly identical (Figure 5A). Although PDCD4 showed an opposite decrease in pancreatic cancer tissues, it was consistent with the trend of MYEOV expression in different stages of pancreatic cancer (Figure 5A). This specificity of PDCD4 might be related to its presence in the nucleus, as studies have found that miRNAs may activate the transcription of related genes in the nucleus [16]. The results of the survival analysis also confirmed the above findings. MYEOV had better OS and DFS at low expression, as did GPRC5A, SERPINB5, KRAS, EGFR, and EIF4G2 (Figure 5B). However, PDCD4 showed the opposite tendency and did not have a significant effect on survival (Figure 5B). This implies that in contrast to the other six genes, PDCD4, although capable of acting as a tumor suppressor gene, may be suppressed in pancreatic cancer [17]. One of the possible reasons may be the overexpression of MYEOV. From the results of the comprehensive analyses of the seven genes in the GSCA database, EIF4G2 showed less variation in different stages. Its differential expression had the least effect on survival (Figure 5C), which may be due to its role as a eukaryotic translation initiation factor that initiates the protein translation process in cells and can be inhibited by PDCD4 binding [18]. We analyzed mRNA expression levels of MYEOV, GPRC5A, KRAS, EGFR, SERPINB5, EIF4G2, and PDCD4 in pancreatic cancer tissues and normal tissues adjacent to pancreatic cancer from three pancreatic cancer patients. It was found that all genes except PDCD4 showed high expression in pancreatic cancer tissues, which was also consistent with our above findings (Figure 5D).

### 3.6. Immune Infiltration Analysis of Genes Associated with the MYEOV Regulatory Network

In pancreatic cancer, we performed a tumor-associated immune cell infiltration analysis for these seven genes together. It was found that PDCD4-associated cell infiltration was almost completely opposite to the other genes, with PDCD4 expression being associated with increased cytotoxic cells, CD4 T cells, γδT cells, MAIT cells, NK cells, Tfh cells, and CD8 T cells infiltration and negatively correlated with monocytes, nTreg cells and Th17 cells (Figure 6A). In contrast, the results analyzed for EGFR, SERPINB5, MYEOV, GPRC5A, and KRAS showed the opposite trend to PDCD4, with relatively increased infiltration of monocytes, nTreg cells, and Th17 cells, while CD4 T cells, γδT cells, MAIT cells, NK cells, Tfh cells, and CD8 T cells significantly decreased (Figure 6A). The increase in EIF4G2-related central memory cells, iTreg cells, and Tr1 cells, with little change in most immune cells, may also account for its lack of significant impact in the pancreatic cancer survival analysis (Figure 5B and Figure 6A). In the same dataset, monocytes, nTreg cells, and Th17 cells were considered significantly upregulated immune cells in pancreatic cancer. In contrast, CD4 T cells, γδT cells, MAIT cells, NK cells, Tfh cells, and CD8T cells were downregulated immune cells in pancreatic cancer (Figure 6B,C). This suggests that EGFR, SERPINB5, MYEOV, GPRC5A, and KRAS may drive further pancreatic cancer development by recruiting pro-tumor-associated immune cells such as monocytes, nTreg cells, and Th17 cells through downstream responses while inhibiting the infiltration of tumor-suppressive immune cells such as CD4 T cells, γδT cells, MAIT cells, NK cells, Tfh cells, and CD8T cells to reduce tumor-associated immune killing [19,20,21,22]. All these results further corroborate and explain our speculation about the role of the molecular network of MYEOV-related genes for pancreatic cancer.

## 4. Discussion

Through bioinformatic mining of genes differentially expressed in normal pancreatic tissue and pancreatic cancer, we identified the survival-related and significantly upregulated gene MYEOV in pancreatic cancer and determined the importance of MYEOV in pancreatic cancer by multiple analytical methods. MYEOV is a myeloma overexpressed gene and the first identified oncogenic human orphan gene [23]. Orphan genes, also known as pedigree restriction genes, are restricted to a specific pedigree and have no similar sequences to other pedigrees [24]. Numerous studies have shown that the incidence of many cancers, such as breast cancer and lung cancer, is less than 2% in hominids, which is much lower than the incidence in humans [25]. This suggests that the orphan gene may be an important factor in the development of human tumors. Although MYEOV has been found to have a promotive effect on pancreatic cancer, it still cannot fully explain the survival-related specificity of MYEOV in pancreatic cancer upregulated genes [26]. 

Through a multiplex analysis approach, we explored that MYEOV has a positively correlated expression relationship with GPRC5A and KRAS, and further developed a molecular network of pancreatic cancer-related genes centered on MYEOV. GPRC5A is expressed as an oncogene in various cancers, and its oncogenic role in pancreatic cancer has also been validated [27,28,29]. Zhou et al. found that GPRC5A can promote pancreatic cancer growth by regulating the gemcitabine-associated tumor-critical mediator HuR [30]. Jahny et al. found that GPRC5A could promote pancreatic cancer development by activating STAT3 [31]. Collins et al. found that GPRC5A was more highly expressed in KRAS-mutated pancreatic cancers, which is consistent with our findings [32]. KRAS has been extensively studied in pancreatic cancer, and almost all studies pointed out its importance in the development of pancreatic cancer [33]. KRAS belongs to the RAS gene family, and its mutations can be observed in more than 90% of PDACs [34]. The protein-encoding KRAS is a GTPase protein, which in cancerous cells is activated to a GTP-bound activation state by cellular transmembrane receptor signaling such as EGFR, which further activates various cancer growth and proliferation pathways such as downstream MAPK and AKT [14]. Thus, KRAS is considered an effective target for pancreatic cancer therapy. SERPINB5 belongs to the serine protease inhibitor superfamily [35]. It can be secreted by pancreatic cancer cells and deposited in the ECM, correlating with poor patient prognosis [36]. SERPINB5 can promote cancer development by participating in TGFβ and fibrosis regulation [37]. EGFR is a prototypical receptor tyrosine kinase and can activate KRAS signaling downstream [38]. EGFR ligands can also promote phosphorylation of SERPINB5 [39]. PDCD4 has shown tumor suppressive effects in a variety of tumors including pancreatic, colon, liver, and breast cancers [17]. PDCD4 can bind to EIF4G2 to inhibit the translation of mRNAs of some growth-promoting genes and oncogenes, which often have structured 5’ untranslated regions [40].

A previous study found that MYEOV does not express proteins in non-small cell lung cancer but acts as a ceRNA to activate the TGF-β pathway to promote the biogenesis of lung cancer [11]. They pointed out that the MYEOV transcript was robustly overexpressed in NSCLC without being translated into a protein product, suggesting that the MYEOV transcript might act as a functional RNA molecule in cells. This is consistent with our results that MYEOV promotes pancreatic cancer primarily through ceRNA function independent of protein-coding capacity. Analyzing the miRNAs bound by MYEOV-related genes, we found that GPRC5A, SERPINB5, EGFR, KRAS, PDCD4, and EIF4G2 all have two miRNAs that act with them, namely hsa-miR-103a-3p and hsa-miR-107. These two miRNAs are also targets for MYEOV interaction. This implies that these pancreatic cancer-related genes, regulated by miRNA-mediated gene regulatory network, can collectively affect the malignant biological behavior of pancreatic cancer. Further, in the future, the corresponding molecular prediction or construction of targeted therapeutic drugs can be performed for MYEOV.

In addition, this study performed a comprehensive immune infiltration analysis of MYEOV-associated genes and found that these genes may be associated with various tumor immune-promoting and suppressing cells. Due to the specificity of the “cold tumor” immune environment in pancreatic cancer, many immunotherapeutic approaches are ineffective [41]. Studies have shown that downstream malignant transformation by activated KRAS can promote the development of an immunosuppressive environment in pancreatic cancer, and silencing KRAS can reverse this altered immune microenvironment [32]. In addition, vaccines associated with pancreatic cancer suggested that this immunosuppression in pancreatic cancer is modifiable [42]. The immune infiltration analysis of genes in the MYEOV-related molecular network in this present study showed that these genes might promote an immunosuppressive phenotype through various tumor-associated immune cells under the regulation of MYEOV. Drugs targeting the MYEOV molecular network can potentially reverse the development of pancreatic cancer by improving the immunosuppressive environment in pancreatic cancer.

Although this study offers many possibilities for the targeted therapy of pancreatic cancer, there are still some limitations. This study did not validate the intermolecular relationships and the specific effects of molecules on pancreatic cancer through experiments. More in-depth studies can be conducted to further validate our findings. Multi-gene experiments could be conducted in the future to explore the inter-molecular and molecular downstream effects.

## 5. Conclusions

Overall, we identified the key position of MYEOV in pancreatic cancer by multiplex analyses and proposed the core ceRNA network of MYEOV for the first time. We also validated the role of ceRNA network genes such as GPRC5A, SERPINB5, KRAS, EGFR, EIF4G2, and PDCD4 in pancreatic cancer. The importance of this molecular network in the mechanism of pancreatic cancer was further confirmed by exploring the effects of these genes on tumor-associated immune infiltrating cells. The new findings of the present study provide more possibilities for early prediction and targeted treatments of pancreatic cancer.

## Figures and Tables

**Figure 1 cancers-14-05439-f001:**
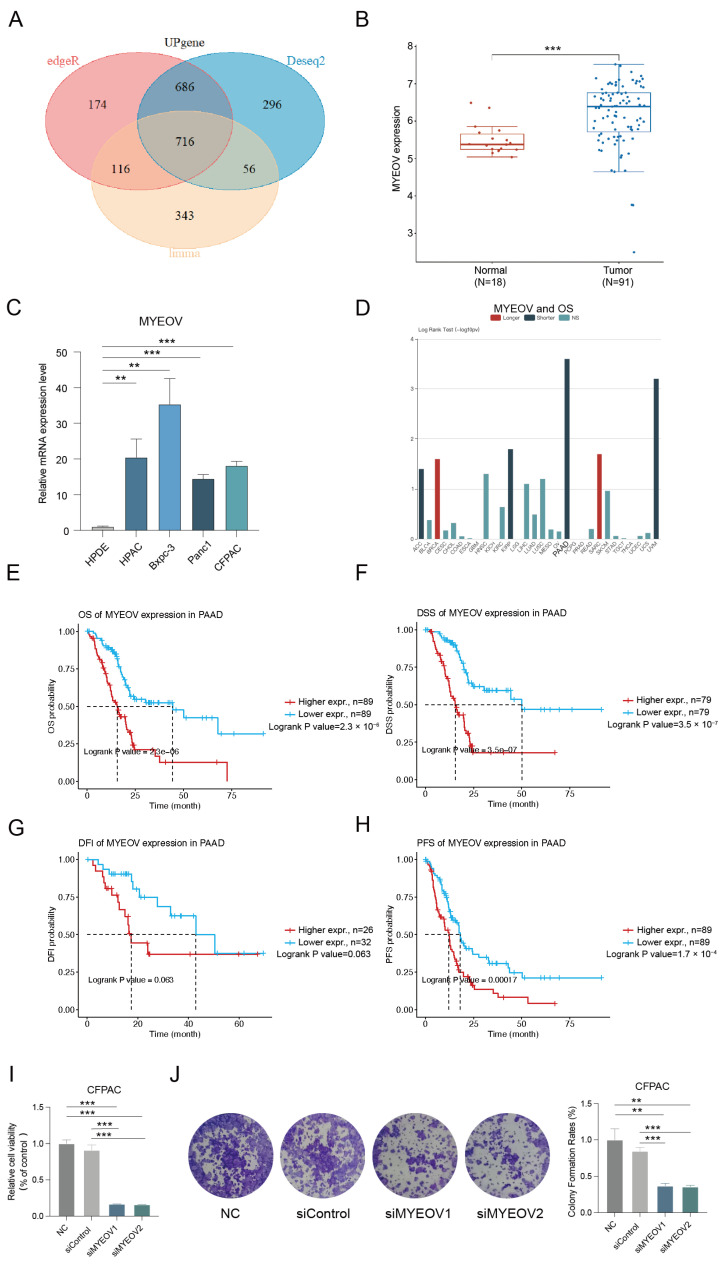
MYEOV was found to be a key upregulated gene associated with pancreatic cancer. (**A**) Intersection was performed to obtain the upregulated genes simultaneously analyzed by three methods. (**B**) Results of differential gene expression analysis combined by GSE107610 and GSE16515 datasets from the GEO database. T, tumor tissue; N, normal tissue. (**C**) Relative mRNA levels of MYEOV in HPDE and some pancreatic cancer cells (HPAC, Bxpc-3, Panc1, CFPAC). N = 3. ** *p* < 0.01, *** *p* < 0.001. (**D**) MYEOV was linked to shorter OS in pancreatic cancer. PAAD, pancreatic adenocarcinoma. (**E**) OS curves of MYEOV expression in PAAD. (**F**) DSS curves of MYEOV expression in PAAD. (**G**) DFI curves of MYEOV expression in PAAD. (**H**) PFS curves of MYEOV expression in PAAD. (**I**) Relative cell viability changes of CFPAC cells after knockdown of MYEOV with siRNAs. (**J**) Relative colony formation rate changes of CFPAC cells after knockdown of MYEOV with siRNAs.

**Figure 2 cancers-14-05439-f002:**
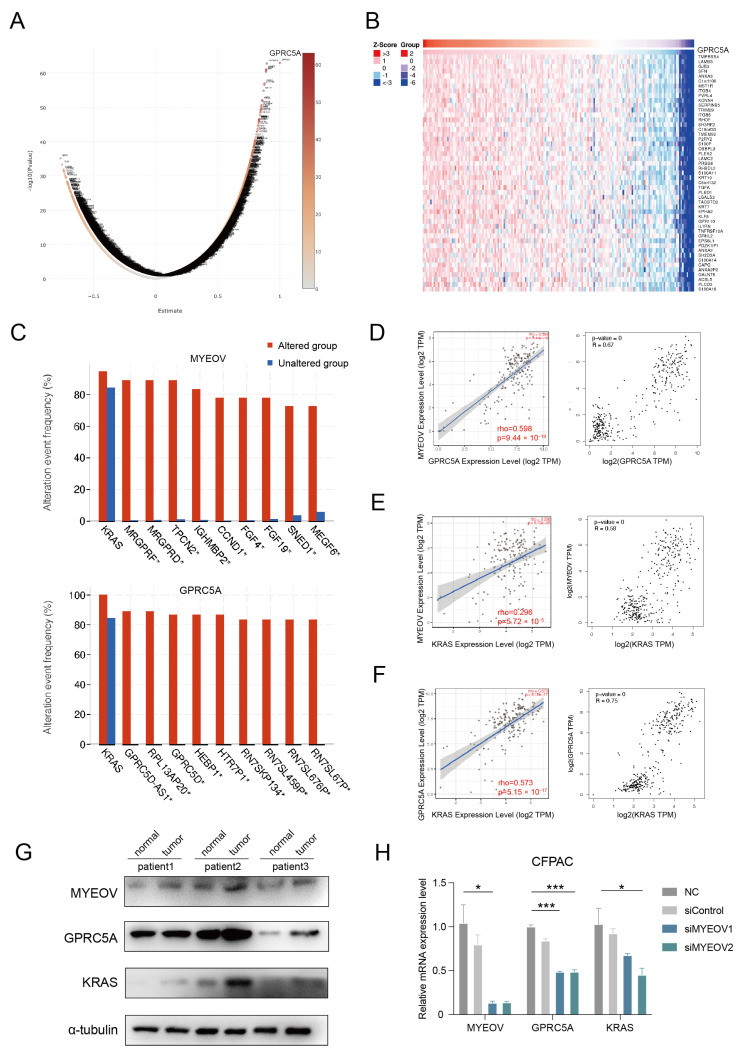
Gene co-expression analysis of MYEOV in pancreatic cancer. (**A**) Volcano map of genes obtained from MYEOV co-expression analysis. (**B**) Heat map of positively correlated genes from MYEOV co-expression analysis. (**C**) Alteration-associated gene predictions for MYEOV and GPRC5A, respectively, by the cBioPortal database. (**D**) There was a strong positive association between MYEOV and GPRC5A. (**E**) There was a strong positive association between MYEOV and KRAS. (**F**) There was a significant positive association between GPRC5A and KRAS. (**G**) Protein expression levels of MYEOV, GPRC5A, and KRAS in pancreatic cancer tissues and normal tissues adjacent to pancreatic cancer from three pancreatic cancer patients were detected by western blotting. (**H**) Changes in mRNA levels of MYEOV, GPRC5A, and KRAS after knockdown of MYEOV with siRNAs in CFPAC cells. * *p* < 0.05, *** *p* < 0.001.

**Figure 3 cancers-14-05439-f003:**
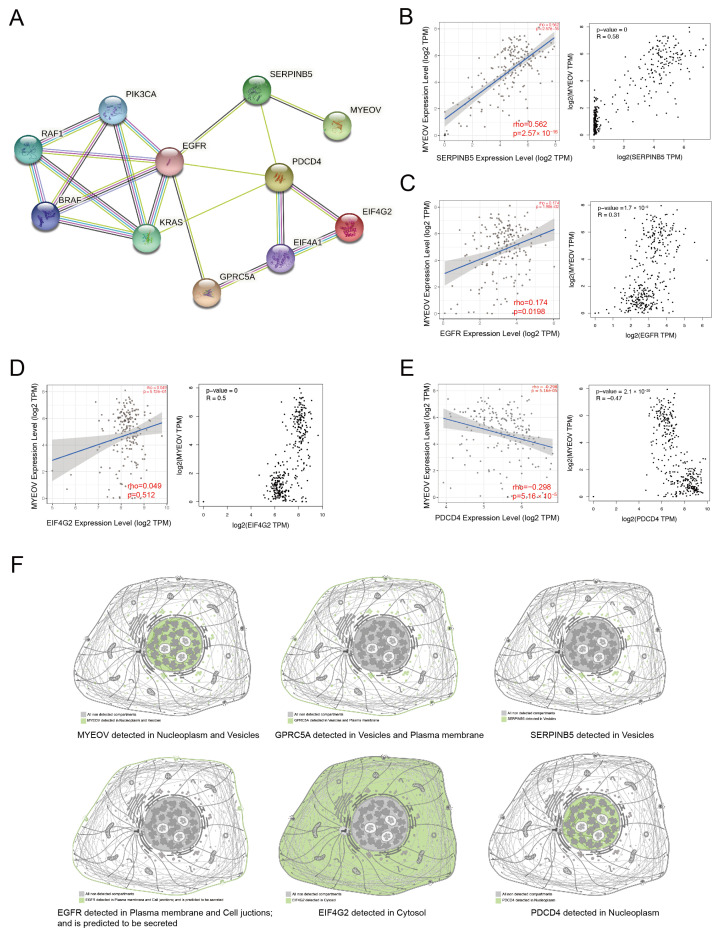
PPI analysis and molecular regulatory network of MYEOV. (**A**) MYEOV, GPRC5A, and KRAS had protein interactions with genes such as SERPINB5, EGFR, EIF4G2, and PDCD4. The light green lines indicate text mining, the black lines indicate co-expression, the purple lines indicate their relationship was experimentally determined, and the blue lines indicate their relationship was from curated databases. (**B**) MYEOV was significantly associated with SERPINB5 by the analysis of the Timer2 database and the GEPIA database. (**C**) MYEOV was significantly associated with EGFR by the analysis of the Timer2 database and the GEPIA database. (**D**) MYEOV was significantly associated with EIF4G2 by the analysis of the Timer2 database and the GEPIA database. (**E**) MYEOV was significantly associated with PDCD4 by the analysis of the Timer2 database and the GEPIA database. (**F**) The subcellular localization of MYEOV, GPRC5A, SERPINB5, EGFR, EIF4G2, and PDCD4 proteins.

**Figure 4 cancers-14-05439-f004:**
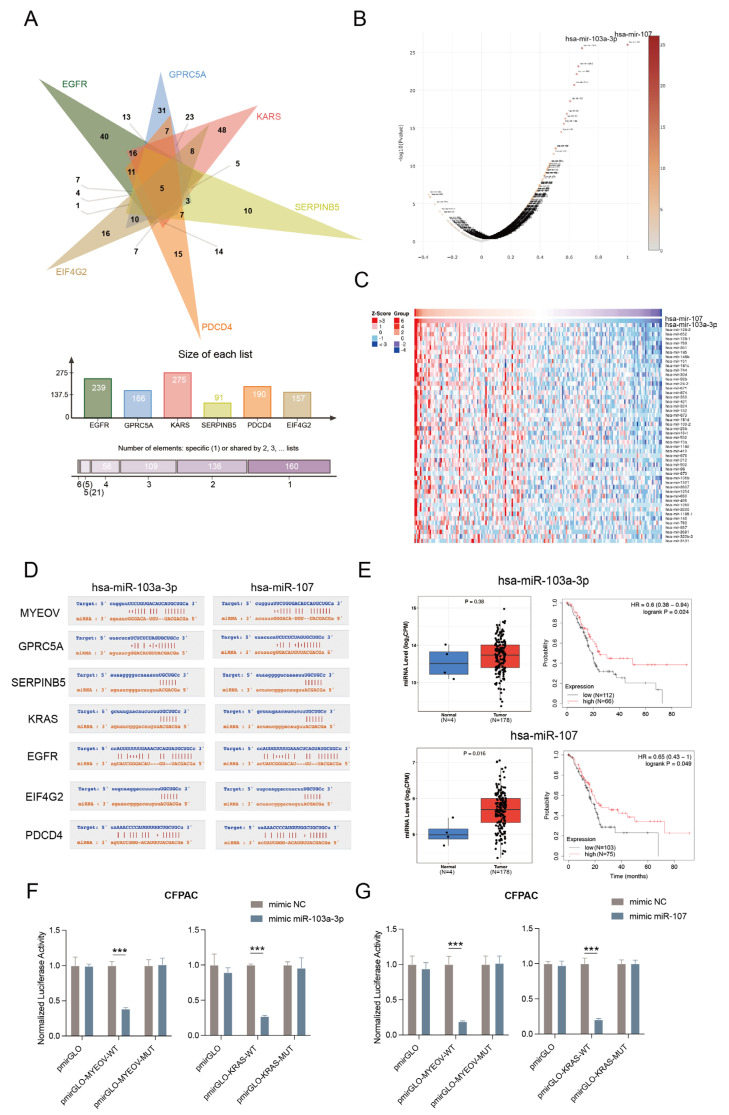
MYEOV acts as ceRNA for the regulation of multiple genes. (**A**) Intersections of miRNAs bound to GPRC5A, SERPINB5, KRAS, EGFR, EIF4G2, and PDCD4. (**B**) Volcano map of miRNAs correlated with hsa-miR-107. (**C**) Heat map of miRNAs positively correlated with hsa-miR-107. (**D**) The active sites of two miRNAs with MYEOV, GPRC5A, SERPINB5, KRAS, EGFR, EIF4G2, and PDCD4. (**E**) The expression analyses and survival curves of hsa-miR-103a-3p and hsa-miR-107. (**F**) Dual-luciferase reporter gene assays confirmed the binding specificity of MYEOV/KRAS and hsa-miR-103a-3p. (**G**) Dual-luciferase reporter gene assays confirmed the binding specificity of MYEOV/KRAS and hsa-miR-107. WT, wild type; MUT, mutation. *** *p* < 0.001.

**Figure 5 cancers-14-05439-f005:**
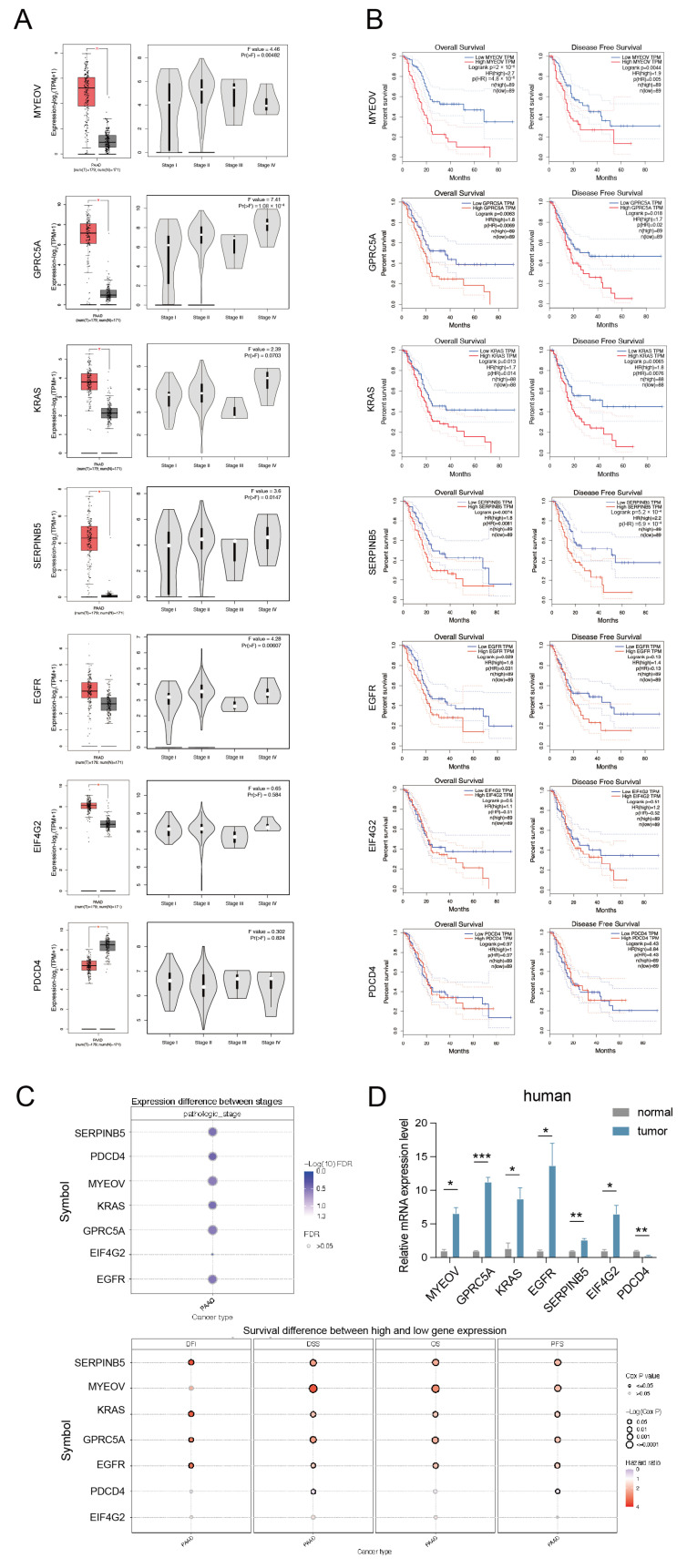
Analysis of the role of related genes in MYEOV regulatory network on pancreatic cancer. (**A**) Differential expression analysis of MYEOV, GPRC5A, SERPINB5, KRAS, EGFR, EIF4G2, and PDCD4 in pancreatic cancer. (**B**) OS and DSS analyses of MYEOV, GPRC5A, SERPINB5, KRAS, EGFR, EIF4G2, and PDCD4 in pancreatic cancer. (**C**) Bubble plots of differential expression analyses with different stages and DFI, DSS, OS, and DFS analyses of MYEOV, GPRC5A, SERPINB5, KRAS, EGFR, EIF4G2, and PDCD4 in pancreatic cancer. (**D**) The mRNA expression levels of MYEOV, GPRC5A, KRAS, EGFR, SERPINB5, EIF4G2, and PDCD4 in pancreatic cancer tissues and normal tissues adjacent to pancreatic cancer from three pancreatic cancer patients. * *p* < 0.05, ** *p* < 0.01, *** *p* < 0.001.

**Figure 6 cancers-14-05439-f006:**
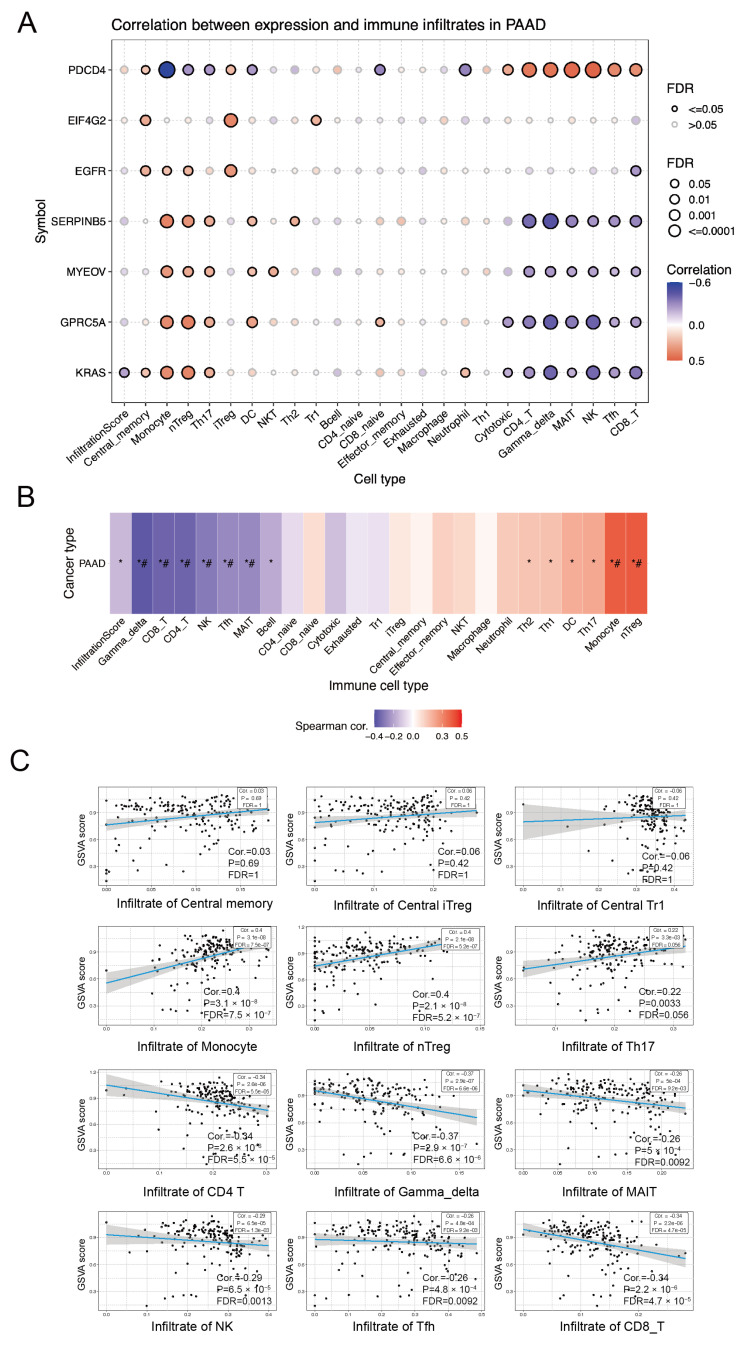
Immune infiltration analysis of genes associated with the MYEOV regulatory network. (**A**) Immune cell infiltration analyses of MYEOV, GPRC5A, SERPINB5, KRAS, EGFR, EIF4G2, and PDCD4 in pancreatic cancer. (**B**) Correlation analysis of infiltrated immune cells and pancreatic cancer. Positive correlation is in red and negative correlation is in blue. * *p* value ≤ 0.05; ^#^ FDR ≤ 0.05. (**C**) Dot maps of spearman correlation between GSVA score and infiltrated immune cells in PAAD.

## Data Availability

The RNA sequencing data were downloaded from the TCGA database (https://cancergenome.nih.gov/abouttcga/overview (accessed on 30 April 2022)).

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
