# Peer review of "Identification of MYEOV-Associated Gene Network as a Potential Therapeutic Target in Pancreatic Cancer"

_cancers, 2022, doi:10.3390/cancers14215439_

Round 1

Reviewer 1 Report (Previous Reviewer 1)

My concerning issues have been improved.

Reviewer 2 Report (Previous Reviewer 2)

 Author has addressed the suggested queries and I recommend to accept the revised manuscript.

Reviewer 3 Report (Previous Reviewer 3)

In the revised version, they have addressed all major concerns however, the new data are not strong.  

Reviewer 4 Report (Previous Reviewer 4)

While the authors have provided satisfactory answers to most of my specific comments none of my main concerns (mentioned in the general evaluation of my review) are addressed at all therefore I am still not supporting the publication of the revised manuscript. 

This manuscript is a resubmission of an earlier submission. The following is a list of the peer review reports and author responses from that submission.

Round 1

Reviewer 1 Report

The slight difference of MYEOV protein level (Fig. 2G) might be able to explain by the author’s response. Meanwhile, the data in Fig.4, especially survival curves of has-miR-107 (Fig. 4E), can not be accepted for publication. There is significant difference of the mRNA level between normal and tumor samples, but not between low and high groups. Previous Fig. 4F could be fine and interesting, but the survival analysis should be performed again after alteration of criteria (change of low/high levels). Moreover, it is so strange that exactly the same pictures are used in the manuscript (Figs. 1 and 5). The data of MYEOV in Fig.5 should be moved and be summarized in Fig.1.

Reviewer 2 Report

The study conducted by Yu Chen entitled “Identification of MYEOV-associated gene network as a potential therapeutic target in pancreatic cancer” identified MYEOV (myeloma overexpressed gene) as significantly overexpressed gene in pancreatic cancers, author also found the association of MYEOV overexpression with overall poor survival in pancreatic cancers. The study has been designed properly. Still, some experiments would help the readers grab the story better. The observation is interesting, but the manuscript is not acceptable in its present form. As per my consideration, it needs some more validation experiments before final acceptance.

Comments

Author has reported that overexpression of MYEOV in pancreatic cancer and the overexpression of MYEOV is associated with poor survival in pancreatic cancer. The association of MYEOV overexpression with pancreatic cancer looks promising and interesting. I recommend testing the effect of MYEOV depletion (knockout or knockdown) on pancreatic cancer cell growth by using cell proliferation and colony forming experiment.

Reviewer 3 Report

The manuscript “Identification of MYEOV-associated gene network as a potential therapeutic target in pancreatic cancer” by Chen et al reports the role of MYEOV in the progression of pancreatic cancer. Authors found that MYEOV mRNA was overexpressed in pancreatic cancers and the level of MYEOV mRNA was strong associated with the survival of patients with pancreatic cancer. They also found that MYEOV could serve as ceRNA to affect the levels of GPRC5A, SERPINB5, EGFR, KRAS, EIF4G2 and PDCD4 by producing sponging effects on hsa-miR-103a-3p and hsa-miR-107. Most results are clearly presented and convincing. Most conclusions are supported by the experimental results. However, there are some concerns, which need to be further addressed as follows.

1. In Figure 2, authors showed changes in mRNA levels of MYEOV, GPRC5A and KRAS after knockdown of MYEOV with siRNAs. Authors should also show whether the protein levels of MYEOV, GPRC5A and KRAS were down-regulated after MYEOV siRNA transfection because the proteins are final modulators of signaling.

2. In Figure 4, authors showed that hsa miR-103a-3p and hsa-miR-107 could bind to MYEOV by computerized analysis. However, authors should conduct transfection experiment to confirm that miR-103a-3p and hsa-miR-107 can biologically bind to MYEOV before they can claim that MYEOV functions as ceRNA.

3. Similarly, authors should conduct transfection experiment to confirm that miR-103a-3p and hsa-miR-107 can biologically bind to GPRC5A, SERPINB5, EGFR, KRAS, EIF4G2 and PDCD4 to confirm the MYEOV-associated gene network authors claimed.

4. In Discussion, author mentioned “our results also indicated that the effect of MYEOV on pancreatic cancer progression could be based on its circRNA form. However, authors didn’t show any results regarding circular MYEOV expression and circMYEOV sponge.

Reviewer 4 Report

In this manuscript, the authors have used an extensive panel of bioinformatics tools and databases to investigate the role of MYOEV in pancreatic cancer. They computationally identified a molecular network which is as stated by the authors to be ‘the molecular mechanism that promotes pancreatic cancer’ and consequently they conclude that the gene MYOEV is 'the cause' of pancreatic cancer.  

While few other studies (DOI: 10.1080/15384101.2020.1757243; doi: 10.1038/s41388-020-01443-4.; doi: 10.1038/s41419-021-04387-z.) have already proven that MYOEV is indeed a prognostic biomarker of PDAC reducing greatly the novelty of a key finding of the manuscript, the greatest limitation of this study is that it does not validate ‘the intermolecular relationships and the specific effects of molecules on pancreatic cancer through experiments’ as mentioned by the authors selves (line 435-436).

Many strong conclusions are made through the manuscript while the data obtained are almost never validated. The western blot in fig 2G is not supportive of the fact that MYOEV is more abundant in tumor material for instance. On the other hand the transient knock down of MYOEV seems to affect slightly the level of GPRC5A and KRAS but it does not prove that all three genes are co-regulated during tumor progression.

Overall, regardless the fact that most of the data are all obtained from databases, the findings are limited in novelty and there is no clear proof that MYOEV does act indeed as a ceRNA as hypothesized by the authors; therefore the manuscript would require extensive improvements to be considered for publication.

Specific remarks:

-           Unclear which hypothesis the authors aim to prove in this study. Is it yes or no the MYOEV mRNA or the protein which is relevant for the cancer cells?

-           The summary of the findings in the intro (from line 70 to 80) is very vague and not informative.

-           In general the English (choice of scientific terms and grammar) should be improved.

-           The material and methods is too succinct and consequently too difficult to eventually reproduce. (what are the three groups of human pancreatic cancer patients in line 97/98)

-           There is no marker used in the western blot to prove that normal and tumor tissue are indeed of different origins. Why did the authors did not include a qRT-PCR for MYOEV of those samples?

-           To show any ‘relationship’ between the three genes, maybe that a pull down could help out but rather more interesting would be to know which common transcription factor regulates the transcription of the three genes.

-           It is quite unclear what the authors aimed at when looking at the intracellular localization using data from the protein atlas. What is the rationale behind this? Are all three genes part of a specific pathway? Even the PPI data do not support any real relationship between the genes.